# A machine learning based exploration of COVID-19 mortality risk

**Mahdi Mahdavi** [1,2☯], **Hadi Choubdar**[1,2☯], **Erfan Zabeh**[3☯], **Michael Rieder**[4,5,6,7],
**Safieddin Safavi-Naeini**[8], **Zsolt Jobbagy**[9], **Amirata Ghorbani**[10], **Atefeh Abedini**[11],
**Arda Kiani**[12], **Vida Khanlarzadeh**[6], **Reza Lashgari**[1], **Ehsan Kamrani**[4,8]*

**1** Institute of Medical Science and Technology (IMSAT), Shahid Beheshti University, Tehran, Iran,
**2** Department of Medicine, Shahid Beheshti University of Medical Sciences, Tehran, Iran, **3** Department of
Biomedical Engineering, Columbia University, New York, NY, United States of America, **4** Robarts Research
Institute, University of Western Ontario, London, ON, Canada, **5** Department of Paediatrics, Children's
Hospital of Western Ontario, London, ON, Canada, **6** Department of Medicine, Schulich School of Medicine
and Dentistry, University of Western Ontario, London, ON, Canada, **7** CIHR-GSK Chair in Pediatric Clinical
Pharmacology, Children's Health Research Institute, London, ON, Canada, **8** CIARS (Centre for Intelligent
Antenna and Radio Systems), Department of Electrical and Computer Engineering, University of Waterloo,
Waterloo, ON, Canada, **9** Department of Pathology, Immunology and Molecular Pathology, Rutgers New
Jersey Medical School, Newark, NJ, United States of America, **10** Department of Electrical Engineering,
Stanford University, Stanford, CA, United States of America, **11** Chronic Respiratory Diseases Research
Center, National Research Institute of Tuberculosis and Lung Diseases (NRITLD), Shahid Beheshti
University of Medical Sciences, Tehran, Iran, **12** Tracheal Diseases Research Center, National Research
Institute of Tuberculosis and Lung Diseases (NRITLD), Shahid Beheshti University of Medical Sciences,
Tehran, Iran

☯ These authors contributed equally to this work.
* ekamrani@uwaterloo.ca

doi.org/10.1371/journal.pone.0252384

Sciences and Technology (NUST), PAKISTAN

**Data Availability Statement:** Data are available in
Figshare. Doi: 10.6084/m9.figshare.14723883
https://doi.org/10.6084/m9.figshare.14723883.

**Funding:** The author(s) received no specific
funding for this work.

## Abstract

Early prediction of patient mortality risks during a pandemic can decrease mortality by
assuring efficient resource allocation and treatment planning. This study aimed to develop
and compare prognosis prediction machine learning models based on invasive laboratory
and noninvasive clinical and demographic data from patients' day of admission. Three Sup-
port Vector Machine (SVM) models were developed and compared using invasive, non-
invasive, and both groups. The results suggested that non-invasive features could provide
mortality predictions that are similar to the invasive and roughly on par with the joint model.
Feature inspection results from SVM-RFE and sparsity analysis displayed that, compared
with the invasive model, the non-invasive model can provide better performances with a
fewer number of features, pointing to the presence of high predictive information contents in
several non-invasive features, including $SPO_2$, age, and cardiovascular disorders. Further-
more, while the invasive model was able to provide better mortality predictions for the immi-
nent future, non-invasive features displayed better performance for more distant expiration
intervals. Early mortality prediction using non-invasive models can give us insights as to
where and with whom to intervene. Combined with novel technologies, such as wireless
wearable devices, these models can create powerful frameworks for various medical
assignments and patient triage.

**Competing interests:** The authors have declared that no competing interests exist.

## Introduction

The SARS-COV-2 pandemic has tremendously strained economic and healthcare infrastructures worldwide, leaving a trail of more than 1.6 million deaths behind as of December 22, 2020 [1]. With no effective treatment and the possibility of emerging new viral strains, an average global death rate of around 6000 per day could lead to the death of approximately 2.2 million individuals in one year. Even though strict social distancing and preventive measures are still in effect, the global mortality and prevalence curve of the disease shows little improvement [1]. More focus on early clinical interventions could be helpful in reducing mortality rates. Critical patients will need timely intensive care unit (ICU) admission and ventilators. In China, it has been reported that about 54% of critical patients were unable to receive timely ICU care, and 30% of patients who died did not receive mechanical ventilation in time [2]. With large patient loads, exhausted medical personnel, and insufficient medical resources, expedited identification of patients that have high mortality risks becomes a key factor in decreasing patient deaths.

Physicians are often unable to accurately predict the prognosis of COVID-19 patients upon their admission until later stages of the disease. Furthermore, the course of COVID-19 can take unpredictable turns where the condition of a seemingly stable patient deteriorates rapidly to a critical state [3]; this could catch even the most skilled physicians off guard. To enhance clinical prediction, Artificial Intelligence (AI) models could be valuable assistants since they can detect complex patterns in large datasets [4, 5]; a capability the human brain is inept at. AI tools have been recruited to fight COVID-19 [6] on various scales, from epidemiological modelling [7, 8] to individualized diagnosis [9, 10] and prognosis prediction [11–14]. Although several COVID-19 prognostic models have been proposed [15], no comprehensive study has evaluated and compared the prognostic prediction power of non-invasive and invasive features.

The aim of this study was three-fold; first, to develop a mortality prediction model from patients' first day of admission routine clinical data; second, to investigate the possibility of predicting COVID-19 mortality outcome using non-invasive patient features; third, to provide a direct comparison of mortality prediction powers between non-invasive and invasive features. Patient data was divided into invasive laboratory tests and non-invasive demographic and clinical features. Three machine learning models were developed to investigate and compare the prediction power of the aforementioned feature groups; two using each of these groups and one using both (Fig 1). It has been reported that many COVID-19 patients experienced their first exacerbation period 24 to 48 hours after admission [16]. Accordingly, we based our model on data from the first day of patients' admission to provide a tool that can be beneficial in real-life scenarios.

## Results

### Data resources

Electronic medical records of 628 patients who were admitted to Masih Daneshvari Hospital between February 20[th], 2020, and May 4[th], 2020, were initially included. Patient diagnosis and severity classification was carried out using the criteria presented in Table 1. After the exclusion of 136 patients, data from 492 individuals (66.1% male, 33.9% female) were used for model development (S1 Fig in S1 File). The median age of the study population was 62 (25). Furthermore, 324 (65.8%) patients were documented as severe, and 168 (34.2%) as non-severe cases. Cough (86.1%), dyspnea (81.3%), and fever (71.4%) were the three most frequent symptoms. Hypertension (38.2%), diabetes mellitus (32.1%), and cardiovascular disease (21.1%)

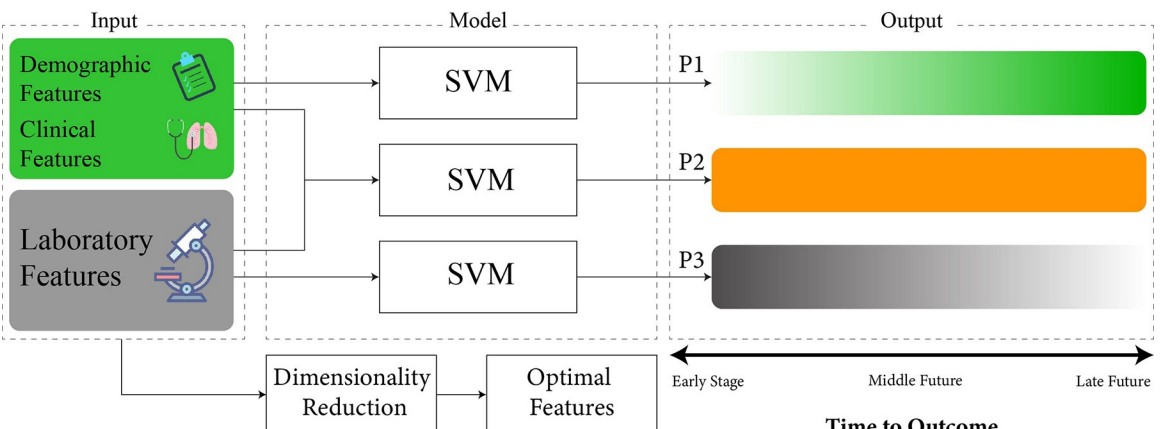

**Fig 1. Illustration of the modeling framework.** Three machine learning models were developed using the SVM framework with three input groups; invasive, non-invasive, and their combination. The invasive group comprises laboratory results. Non-invasive features comprise patient clinical and demographic data. The joint group comprises the combination of invasive and non-invasive features. P1, P2, and P3 represent the prediction performance provided by the non-invasive, joint, and invasive models, respectively. The non-invasive model displayed good prediction performance in the farther future (P1) whereas the invasive model showed good prediction performance for the near future (P3). Neighborhood Component Analysis (NCA), recursive feature elimination via Support Vector Machine (SVM-RFE), and linear SVM with least absolute shrinkage and selection operator (Lasso) sparsity regularization (Sparse Linear SVM) were utilized for inspection of feature contributions and dynamics with respect to the outcome.

**Table 1. Criteria for disease diagnosis and severity assessment upon hospital admission.**

| A. Diagnosis Criteria | | |
| --- | --- | --- |
| **a. Suspected Case** | **b. Probable Case** | **c. Confirmed Case** |
| **1- Fever and/or respiratory symptoms with:** | 1- Suspected case with radiologic features suggestive for COVID infection (multilobular infiltration especially in peripheral areas in CXR or Chest CT scan, Ground Glass Opacity in Chest CT scan) | 1- The presence of SARS-CoV-2 nucleic acid is confirmed in respiratory or blood samples; detected by RT-PCR |
| **• Contact with probable/ confirmed COVID-19 patients within 14 days before the onset** | | |
| **• Healthcare workers** | | |
| **2- Dry cough or chills or sore throat with or without fever** | 2- Suspected patient with pneumonia that is unresponsive to typical medications (clinically confirmed) | |
| | 3- Inconclusive result: a suspected case with unknown PCR test | |
| **B. Patient Severity Classification Criteria** | | |
| **a. Non-Severe** | | **b. Severe** |
| Confirmed COVID infection + Both of the following | | Confirmed COVID infection + One of the following |
| **1- SpO$_2$ $\geq$ 90** | | 1- ICU Admission |
| **2- RR < 30** | | 2- SpO$_2$ < 90 |
| | | 3- RR $\geq$ 30 |

The criteria of this table were utilized by Iranian physicians for diagnosis (A) and severity classification (B) of COVID-19 patients at the time of the data gathering phase. Only confirmed COVID-19 cases were included in the study. CXR: Chest X-Ray, RT-PCR: Real Time-Polymerase Chain Reaction, SpO$_2$: Saturation of Peripheral Oxygen, RR: Respiratory Rate, ICU: Intensive Care Unit.

**Table 2. Demographic, clinical features and mortality outcome of patients collected from medical records.**

| Characteristics | Non-Severe | Severe | Total | P |
|---|---|---|---|---|
| **Age, median (Q$_3$-Q$_1$)** | 57.5 (24.75) | 65.5 (26) | 62 (25) | <0.001 |
| **Sex** | | | | 0.065 |
| Male (%) | 66.1 | 57.4 | 60.4 | |
| Female (%) | 33.9 | 42.6 | 39.6 | |
| **Clinical symptoms on admission** | | | | |
| Cough (%) | 83.6 | 90.3 | 86.1 | 0.11 |
| Fever (%) | 68.1 | 75.3 | 71.4 | 0.48 |
| Fatigue (%) | 35.3 | 40.7 | 38.7 | 0.72 |
| Dyspnea (%) | 72.6 | 92.8 | 81.3 | 0.001 |
| Myalgia (%) | 40.1 | 47.3 | 44.2 | 0.64 |
| **Comorbidities** | | | | |
| Diabetes Mellitus (%) | 28 | 34.3 | 32.1 | 0.186 |
| Hypertension (%) | 30.4 | 42.3 | 38.2 | 0.011 |
| Cardiovascular Disease (%) | 16.1 | 23.8 | 21.1 | 0.049 |
| **Vital Signs** | | | | |
| Blood Pressure max, median (Q$_3$-Q$_1$) | 120 (20) | 120 (25) | 120 (21) | 0.149 |
| Blood Pressure min, median (Q$_3$-Q$_1$) | 70 (10) | 75 (11) | 75 (10) | 0.229 |
| Pulse Rate, median (Q$_3$-Q$_1$) | 88 (16) | 90 (21) | 90 (20) | 0.01 |
| Respiratory Rate, median (Q$_3$-Q$_1$) | 19 (2) | 20 (6) | 20 (5) | <0.001 |
| Temperature, median (Q$_3$-Q$_1$) | 37.5 (0.8) | 37 (0.7) | 37 (0.7) | 0.104 |
| SpO$_2$, median (Q$_3$-Q$_1$) | 93 (4) | 84.5 (13) | 88 (12) | <0.001 |
| **Outcome** | | | | <0.001 |
| Discharge (%) | 95.2 | 45.1 | 62.2 | |
| Expired (%) | 4.8 | 54.9 | 37.8 | |

Data were first tested for normality by the Kolmogorov-Smirnov test. A test level of $\alpha$ = 0.05 and $P < 0.05$ showed that the sample distribution is not normal. Continuous normally distributed variables are described by mean and standard deviation, and continuous non-normally distributed variables are described by median and quartiles.

were the three most frequent comorbidities among patients (Table 2). Descriptive characteristics of laboratory features are displayed in Table 3.

## Contribution of individual factors to mortality prediction

Preliminary analysis of single parameters and earlier reports of COVID-19 mortality prediction [15], suggested that mortality outcome can be predicted by the analysis of individual biomarkers. In this study, an approach from simple to more complex analysis was adapted. Initial analysis and comparison of the contribution of individual features to the prediction of mortality outcome was implemented via Neighborhood Component Analysis (NCA) (Fig 2A). The results of NCA analysis on 37 biomarkers in Fig 2 panel A demonstrated that several non-invasive features (green shaded), such as SPO$_2$ and age, and laboratory biomarkers (gray shaded), such as BUN and LDH, had significant weights in mortality prediction. Outcome prediction using a single biomarker is not highly accurate since a feature's information content is limited and distinct features have different information contents. Fig 2B and 2C illustrate this fact; while data points are roughly visually separable in the feature subspace of Fig 2B (higher amount of information), they are significantly crunched and inseparable in the feature subspace of Fig 2C (lower amount of information). Results from the NCA dimensionality analysis were further evaluated using a Generalized Linear model with the least absolute shrinkage and

**Table 3. Patients' laboratory data collected from medical records.**

| Characteristics | Non-Severe | Severe | Total | P |
|---|---|---|---|---|
| **Complete Blood Count (CBC)** | | | | |
| WBC, median ($Q_3$-$Q_1$) | 6.65 (5.28) | 8.57 (6.87) | 7.8 (6.4) | <0.001 |
| Neutrophil, median ($Q_3$-$Q_1$) | 4.97 (4.4) | 6.83 (6.63) | 6.11 (6.14) | <0.001 |
| Lymphocyte, median ($Q_3$-$Q_1$) | 1.35 (0.93) | 1.16 (0.77) | 1.22 (0.88) | <0.001 |
| RBC, median ($Q_3$-$Q_1$) | 4.51 (0.92) | 4.44 (1.07) | 4.46 (0.97) | 0.737 |
| HB, median ($Q_3$-$Q_1$) | 13.2 (3.28) | 13.1 (2.9) | 13.1 (3.1) | 0.348 |
| HCT, median ($Q_3$-$Q_1$) | 38.45 (7.55) | 38.45 (7.9) | 38.4 (7.8) | 0.772 |
| PLT, median ($Q_3$-$Q_1$) | 187 (81.25) | 186.5 (115) | 186.5 (103) | 0.85 |
| MCV, median ($Q_3$-$Q_1$) | 84.62 (7.59) | 85.92 (7.67) | 85.4 (7.7) | 0.035 |
| MCH, median ($Q_3$-$Q_1$) | 29.29 (3.28) | 29.2 (3.1) | 29.2 (3.2) | 0.326 |
| MCHC, median ($Q_3$-$Q_1$) | 34.31 (2.24) | 33.73 (2.38) | 34 (2.36) | <0.001 |
| RDW, median ($Q_3$-$Q_1$) | 13.4 (2.33) | 14.4 (2.6) | 14 (2.7) | <0.001 |
| ESR, median ($Q_3$-$Q_1$) | 35 (44.75) | 49 (48.75) | 44.5 (47.4) | <0.001 |
| **Coagulation** | | | | |
| PT, median ($Q_3$-$Q_1$) | 13 (1) | 13.7 (2) | 13.4 (1.5) | <0.001 |
| PTT, median ($Q_3$-$Q_1$) | 35 (10) | 39 (15) | 38 (13) | <0.001 |
| INR, median ($Q_3$-$Q_1$) | 1.1 (0.2) | 1.2 (0.42) | 1.15 (0.35) | <0.001 |
| **Biochemistry** | | | | |
| BUN, median ($Q_3$-$Q_1$) | 14.48 (10.24) | 19 (16.98) | 17.14 (16.1) | <0.001 |
| Cr, median ($Q_3$-$Q_1$) | 1.19 (0.41) | 1.2 (0.6) | 1.2 (0.5) | 0.145 |
| Uric Acid, median ($Q_3$-$Q_1$) | 3.1 (2.2) | 4.7 (3.5) | 4.3 (3.9) | 0.012 |
| AST, median ($Q_3$-$Q_1$) | 32 (24.9) | 41.15 (30.8) | 38 (30) | <0.001 |
| ALT, median ($Q_3$-$Q_1$) | 26.05 (29.15) | 31 (31) | 29 (30) | 0.053 |
| ALKp, median ($Q_3$-$Q_1$) | 183 (111) | 201.5 (109.75) | 193 (112) | 0.09 |
| LDH, median ($Q_3$-$Q_1$) | 480 (226.2) | 612 (340.55) | 565 (302) | <0.001 |
| **Blood Gas Test** | | | | |
| *PH*, median ($Q_3$-$Q_1$) | 7.37 (0.08) | 7.36 (0.1) | 7.37 (0.09) | 0.104 |
| *PCO2*, median ($Q_3$-$Q_1$) | 42.2 (9.52) | 43.05 (11.88) | 42.9 (11.1) | 0.282 |
| *PO2*, median ($Q_3$-$Q_1$) | 28.09 (13.7) | 31.5 (19.88) | 29.95 (17.03) | 0.007 |
| *HCO3*, median ($Q_3$-$Q_1$) | 24.8 (4.6) | 24.8 (6) | 24.8 (5.4) | 0.956 |
| *BE*, median ($Q_3$-$Q_1$) | -0.1 (3.91) | -0.02 (5.58) | -0.02 (4.6) | 0.687 |

Data were first tested for normality by the Kolmogorov-Smirnov test. A test level of $\alpha = 0.05$ and $P < 0.05$ showed that the sample distribution is not normal. Continuous normally distributed variables are described by mean and standard deviation, and continuous non-normally distributed variables are described by median and quartiles. WBC: White Blood Cell, RBC: Red Blood Cell, HB: Hemoglobin, HCT: Hematocrit, PLT: Platelet, MCV: Mean corpuscular volume, MCH: Mean corpuscular Hemoglobin, MCHC: Mean corpuscular Hemoglobin Concentration, RDW: Red cell Distribution Width, ESR: Erythrocyte Sedimentation Rate, PT: Prothrombin Time, PTT: Partial Thromboplastin Time, INR: International Normalized Ratio, BUN: Blood Urea Nitrogen, Cr: Creatinine, AST: Aspartate Transaminase, ALT: Alanine Aminotransferase, ALKp: Alkaline Phosphatase, LDH: Lactate Dehydrogenase, *BE*: *Base Excess*

selection operator (Lasso) regularization (LassoGlm). Outcomes of the LassoGlm analysis confirmed that, indeed, several non-invasive and invasive features had prominent weights in mortality prediction (S3 Fig in S1 File). The presence of non-invasive and invasive features with significant prediction weights in dimensionality analyses motivated us to further investigate the possibility of mortality prediction using non-invasive and invasive biomarker groups.

## Predictive capability of invasive and non-invasive models

Following the results of the dimensionality reduction analysis, we explored the possibility of accurate mortality prediction via non-invasive and invasive biomarkers and their

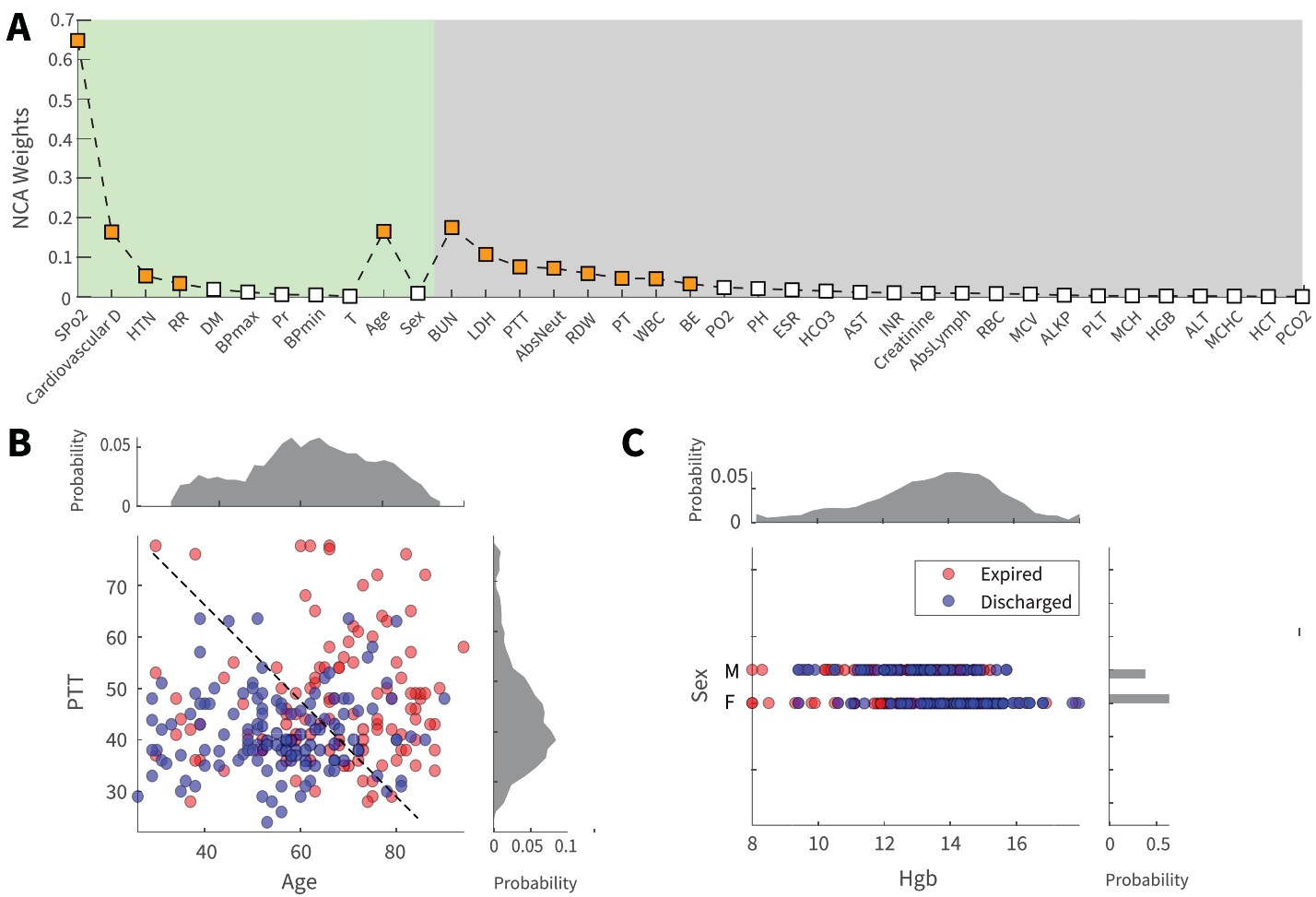

**Fig 2. Contribution of demographic, clinical, and laboratory features to mortality prediction.** (**A**) The results of the regularized NCA analysis displays the contribution of single features to mortality prediction. Features are sorted based on contribution importance and category. Features with prominent weights were displayed by orange squares for visual convenience. (**B**) is a favorable feature space (PTT and age) where the information content of features with respect to the outcome is high, so many data points could be visually distinguished via an illustrative decision border. Panel (**C**), in contrast, demonstrates unfavorable feature space where the low information content of features has led to data points becoming crunched and hard to distinguish (Sex and Hgb). Panels **B** and **C** were created using half of the data and Principal Component Analysis (PCA) for illustrative purposes.

combination. Towards this aim, mortality prediction was implemented using three models (Fig 1). The joint model utilized all the demographic, laboratory, and clinical biomarkers as inputs; this model is considered ideal, where all the required biomarkers are present for prediction. To investigate the differences between invasive and non-invasive features for outcome prediction, two separate models were developed; one solely based on laboratory features (shown by grey color in Figures) and the other only based on non-invasive features (shown by green color in Figures). For each of the Joint, invasive, and non-invasive models, a linear support vector machine (SVM) algorithm was trained and evaluated.

To generalize the results of model prediction to an independent dataset, the data were divided into training and test set. For model training and tuning, 10-fold cross-validation was utilized on instances of the training set [17]. To better validate the performance of the predictive algorithms, independent of the algorithm decision criteria, receiver operating characteristic (ROC) curves were generated. Fig 3A suggested that the prediction performance of the

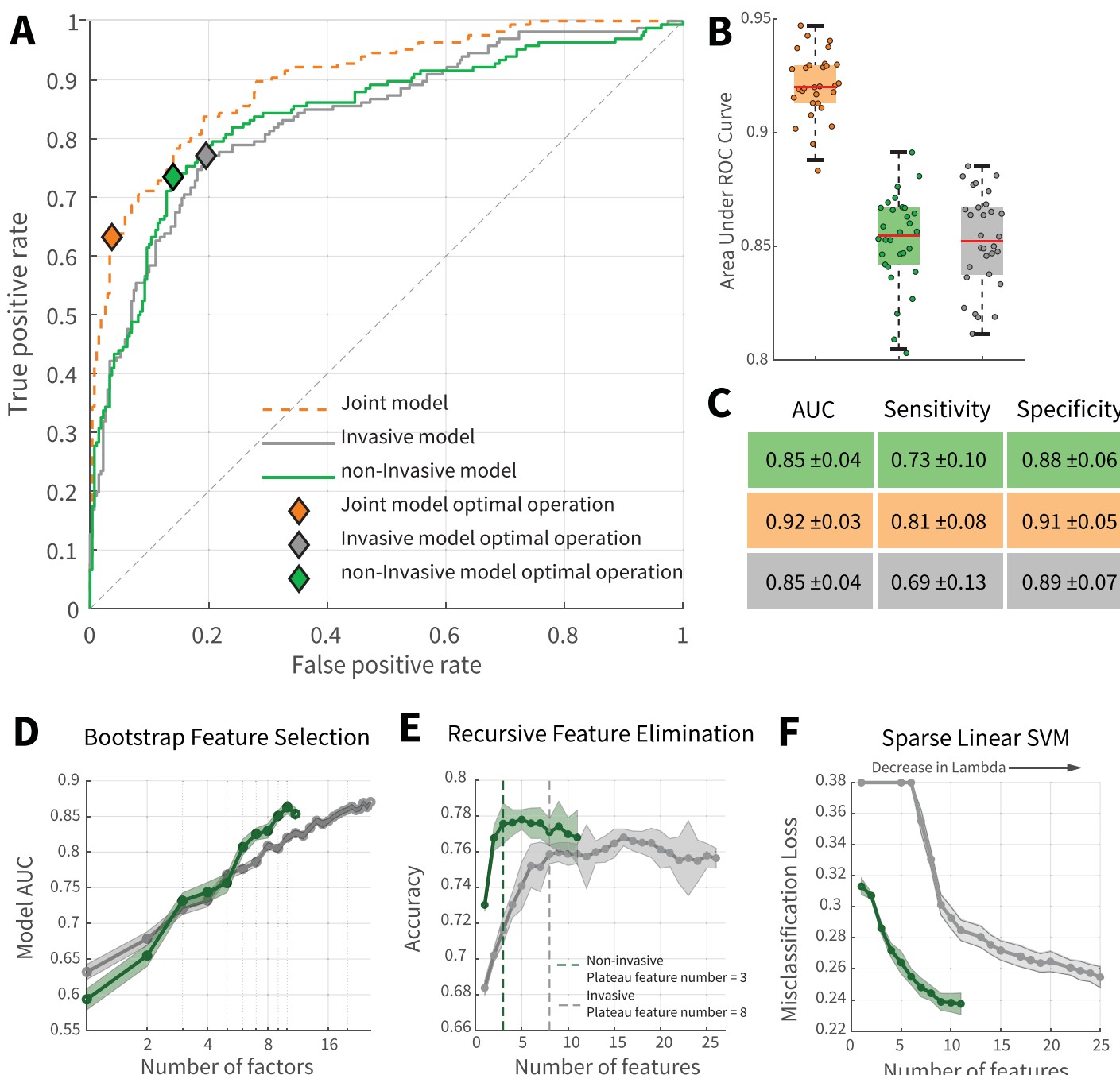

**Fig 3. Comparison of mortality prediction of invasive and non-invasive models.** (**A**) ROC curve of joint, invasive, and non-invasive models. (**B**) Investigation of models' performance and robustness towards sample size. For each data point, a model was trained and evaluated using 90% of data which was randomly bootstrapped from the main dataset while maintaining the original discharge to expired ratio. The models were robust to the sample size and no significant difference was observed between the performance of invasive and non-invasive models. (**C**) Performance table of invasive, non-invasive, and joint models. Performances are reported as mean along with standard deviations. (**D**) Comparing the dynamics of laboratory and non-invasive features for randomly selected combinations of features. (**E**) Recursive feature elimination. Compared with invasive features, prominent non-invasive features had significant prediction information contents. In general, the first three features with prominent contributions to the improvement of the non-invasive model's performance were $SPO_2$, age, and presence of cardiovascular disorders; the first three invasive features were BUN, LDH, and PTT. (**F**) Sparsity analysis. Sparse linear SVM was utilized to investigate optimal feature combinations for fixed predictor numbers. For a specific sparsity level (features number), the non-invasive model performs better than the invasive model. Green and gray represent non-invasive and invasive modes, respectively.

non-invasive model is comparable with the joint model. Although the performance of the non-invasive model was slightly better than the invasive model in Fig 3A, further statistical analysis revealed no significant differences between these models (Fig 3B and 3C); to see if there were significant differences between the performance of models and also test the robustness of data towards the sample volume, 10% hold out cross-validation was performed over 20 iterations. In each iteration, 10% of the data were randomly removed then model training and evaluation were carried out on the remaining data. The results displayed the robustness of models towards the sample size and insignificant difference between invasive and non-invasive models (Fig 3B). The test accuracy of the joint, non-invasive, and invasive models were $0.80 \pm 0.03$, $0.77 \pm 0.04$, and $0.75 \pm 0.4$, respectively. To further evaluate the prediction performance of invasive and non-invasive models, a different classification framework was implemented using an ensemble model of decision trees with Adaptive Logistic Boosting (S4 Fig in S1 File). The results displayed that, indeed, the non-invasive model could achieve performances roughly on par with the joint model (for more details, see the "Ensemble model" in the supplementary section). Therefore, the non-invasive model can be an optimal choice to be implemented and expanded as an assistive triage tool for sieving patients with high mortality risks as it bypasses the high cost and response time of invasive laboratory tests.

## Prediction dynamics of invasive and non-invasive features

Earlier studies of COVID-19 mortality predictions mostly focused on invasive biomarkers [11, 13]. Evaluating invasive biomarkers provides more direct and causal inferences about our physiological state. In contrast, non-invasive features contain broader, indirect information about the body. Thus, it could be hypothesized that accurate mortality risk anticipation is plausible with a sufficient number of non-invasive features. On this basis, we investigated whether the predictive power of non-invasive features (population signal) is comparable to those of invasive features (causal signals). Although the ROC analysis tackles this question, it is still unclear whether the significant performance of the non-invasive model, despite having a lower feature number, is due to the higher absolute information that individual non-invasive measurements carry, or the specific combination of the current measurements. To address this question, we performed three groups of analysis; random bootstrapping, recursive feature elimination via Support Vector Machine (SVM-RFE), and sparsity analysis with linear SVM and Lasso regularization (i.e., sparse linear SVM).

To inspect the dependency of expiration risk information carried with each predictor, in 50 repetitions, the models were trained and compared over a fixed number of predictors which were randomly bootstrapped from the complete set of invasive and non-invasive pools (total iterations = $50^*26 + 50^*11$ iteration). The outcome of this preliminary analysis (Fig 3D) demonstrates the slight superiority of the non-invasive model over the invasive model with the increase of fixed feature numbers. Considering the fact that the model predictors were bootstrapped from full invasive and non-invasive feature spectrum without tuned selection, it can be argued that, compared with invasive features, mortality prediction information of non-invasive features is more independent and several major contributing features are present among them. To further evaluate feature contributions, SVM-RFE was implemented. This framework recursively eliminates features with the lowest prediction weights using an SVM model. The results of SVM-RFE analysis displayed that, indeed, the non-invasive model was able to reach a performance plateau with a lower number of features due to the significant contribution of several individual features. In other words, several non-invasive features had significantly high information content for mortality outcome prediction (Fig 3E). In contrast, a more disperse distribution of information was present among invasive features since,

compared to the non-invasive model, a higher number of invasive features were required for the model to reach a performance plateau.

The previous two methods can have several limitations. For instance, some features might be informative for outcome prediction when they are combined with specific features. Consequently, when the features are recursively removed or randomly picked, these dependent features can become less important for the model if their associated features are absent. To address this issue and further investigate feature dynamics, a sparse linear SVM model with Lasso regularization was adapted for feature inspection. As the strength of the regularization, determined by lambda($\lambda$) variable, increases, more feature coefficients are pushed to zero by Lasso; this induces a state of sparsity where only feature combinations whose contributions are significant for the model are kept. The results showed that with the increase of lambda, the number of prime predictors (i.e., features with non-zero coefficients) of invasive and non-invasive models is reduced. Furthermore, the non-invasive model maintains its superiority over the invasive model in lower feature dimensions. This further supports the hypothesis that several non-invasive features of this study contain a significant amount of predictive information since even in a highly regularized, sparse state, the non-invasive model is able to provide better performances compared with the invasive model (Fig 3F).

Three non-invasive features had substantial contributions towards mortality prediction; these were $SPO_2$, age, and presence of cardiovascular disorders. Among invasive features, BUN, LDH, and PTT were the top three features with the highest prediction contributions. The presence of the aforementioned features as significant predictors was consistent across NCA (Fig 2A), SVM (S2 Fig in S1 File), SVM-RFE (Fig 3E), sparse linear SVM (Fig 3F), LassoGlm (S3 Fig in S1 File), and the ensemble (S5 Fig in S1 File) analyses.

## Comparison of the prediction horizon

According to the functional dissimilarities that we observed between invasive and non-invasive models, we aimed to investigate whether the temporal range of mortality prediction differed between models. The visual inspection of expiration intervals indicated that most patients died within the first week of admission (median = 7 days) with a peak at 3 days (Fig 4A). Next, the data was divided into 8 distinct time intervals based on the expiration date of

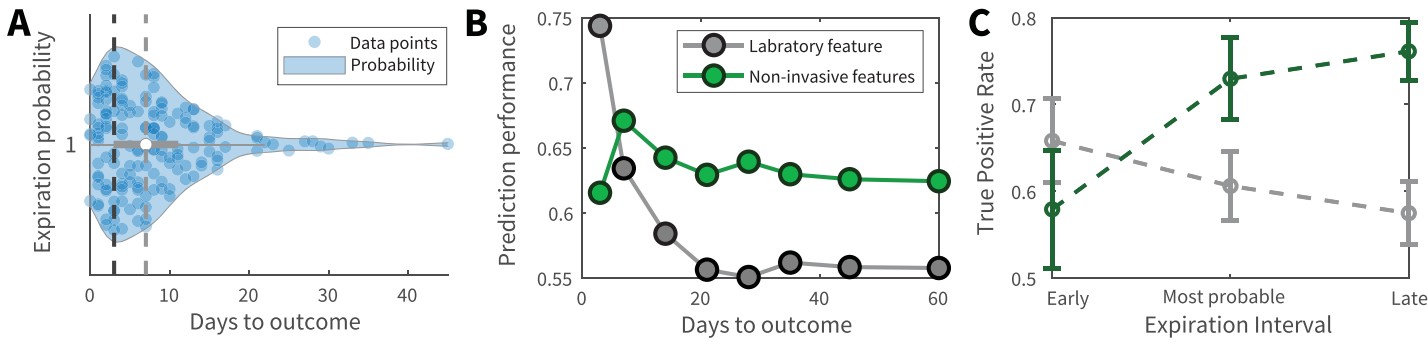

**Fig 4. Temporal range model predictions.** (**A**) Temporal distribution of patient expiration intervals. The black vertical dashed line corresponds to the peak of the expiration distribution which was 3 days from admission. The gray vertical dashed line corresponds to the median expiration interval which was 7 days after admission. (**B**) and (**C**) Prediction performance of invasive and non-invasive models across expiration temporal spectrum. For panel (**B**), invasive and non-invasive models were trained over all the dataset. Afterwards, the expiration prediction performance was evaluated for 8 different expiration intervals. Days to outcome represents the number of days between patient admission and expiration. For panel (**C**), patient data were divided into three expiration intervals; from admission to day 3, from day 3 to day 7, and after day 7. For each interval, independent SVM models were trained and the true expiration ratio (True positive rate) was reported for each interval's model. While invasive features were better predictors for imminent expiration, they were outperformed by non-invasive features over larger expiration intervals. Green and gray represent non-invasive and invasive modes, respectively.

patients. Previous invasive and non-invasive models, each trained over the complete dataset, were utilized to predict patient mortality of these intervals. In a real-word triage scenario, physicians face patients whose outcomes and expiration dates are not known. Therefore, past experiences (here the model trained on all instances) are used for decision making (here predicting expiration of patients from each interval). The fraction of false death prediction to the total number of predictions within each of the days to outcome intervals ($\frac{F}{T+F}$) was calculated and prediction performance was defined as $1 - \frac{F}{T+F}$. This framework tests the models that were trained on patients with various expiration times to see how they would fare in predicting patient mortality across a temporal range.

The comparison between green (non-invasive) and gray (invasive) traces in Fig 4B suggested that while invasive features are more predictive within the last days prior to expiration, they are outperformed by non-invasive features over longer expiration intervals. To further evaluate these results, patient data was divided into 3 expiration intervals; from admission to day 3 (the peak), from day 3 to day 7 (the median), and beyond day 7. Invasive and non-invasive models were trained and evaluated on each interval separately. The results of this analysis further confirmed that, indeed, while invasive features are better predictors for early expiration, they are surpassed by non-invasive features as the distance between admission and time of death increases (Fig 4C).

## Discussion

Conducted for the first time on Iranian patients, this study provided and compared three practical prognostic models using invasive and non-invasive data from the first day of patients' admission to predict the COVID19 mortality. Furthermore, the prediction power of non-invasive and invasive feature groups was evaluated across the temporal and feature number spectrum to reveal interesting results. Compared with the invasive model, the non-invasive model provided better performances in lower, sparse feature dimensions, pointing to the presence of a significant concentration of prediction information in several non-invasive features. In contrast, a more disperse distribution of prediction information was observed among invasive features. Furthermore, while invasive features were good predictors for imminent expiration, they were outperformed by non-invasive features for a more distant expiration interval.

Predicting the trajectory destination of COVID-19 could provide substantial support for decreasing mortality rates. In a pandemic, rapid disease transmission and high patient load could quickly overload healthcare infrastructures; an overloaded medical system can result in higher mortality rates due to inefficient management of limited medical resources; this issue was highlighted by a study indicating that 30% of Chinese COVID-19 patients died without receiving ventilator support [2]. Furthermore, strict preventive measures, social isolation, and pandemic distress could lead to the activation of psychological defensive behaviors in patients where they underestimate their symptoms and do not seek immediate medical assistance [18]. This optimistic bias could be fatal if a patient's condition suddenly worsens towards a critical stage. The disease has an unpredictable trajectory where the condition of some patients suddenly becomes critical [3], surprising even the most skilled physicians; this hampers physicians' performance by limiting their action time window. Moreover, rapid isolation of patients with high mortality risks is required since these patients carry significantly more viral loads even before their condition becomes critical [19]. Similar to an early warning system, our models could be a step in alleviating these problems by providing unbiased, rapid prognosis prediction to support resource allocation and decision making.

We developed three predictive models using invasive features, non-invasive features, and both. Our joint model provides rapid, accurate predictions using features that are routinely

collected upon patient admission, making it implementable even in conditions where imaging or sophisticated laboratory equipment is unavailable. Our results revealed that non-invasive features displayed an overall good prediction capacity compared with the invasive and joint model (Fig 3A). The distribution of predictive information was dispersed among invasive features and they were better predictors for patient expirations in near future. In contrast, several non-invasive features had significant information concentrations and these features were better predictors for deaths that happened further from the admission day (Figs 3 and 4). This difference in prediction dynamics and range might stem from the fact that invasive and non-invasive biomarkers have distinct temporal dynamicity and biological information content. Many key laboratory features, such as LDH and PTT, have high temporal dynamicity; to maintain homeostasis, after an insulting event, these features tend to rise and then return to their normal range in a relatively short time [20]. Furthermore, laboratory feature abnormalities only show disruptions in body systems that they are linked to, limiting their information content. The aforementioned factors require laboratory models to have high feature numbers for accurate prediction and limit their prediction temporal range. In contrast, many non-invasive features, such as age or presence of comorbidity, can be seen as signals that contain a significant amount of compressed, less variable data. Consequently, mortality prediction could be achieved by them with lower dimensions.

Although laboratory features provide valuable information, their analysis requires invasive sampling. Many patients are wary of blood sampling [21]. Moreover, high patient load and equipment shortage could hinder the availability and accuracy of blood testing [22]; many laboratory biomarkers, such as LDH and blood gas tests, require careful sampling, preservation, and transportation to avoid errors from complications, such as hemolysis [23]. Lab tests are also generally expensive. A study from the United States indicates that, even in the absence of a pandemic state, over 20% of patient medical care was not needed [24]. These unnecessary cares will impose a significant financial burden on patients and healthcare systems. Rapid triage of patients is also a critical factor, required to manage high patient loads [25]. However, an important downside of routine rapid triage in a pandemic situation is the increased mortality rate due to missing high-risk patients [14]. These patients might incorrectly be identified as mild and, without further workup, be advised to take a home-treatment approach. Our model using non-invasive features could provide rapid, accurate prognosis prediction without additional costs or waiting time to augment the initial triage and avoid missing high-risk patients. To further automate this triage approach, vital signs could be effortlessly measured and relayed by wireless wearable medical devices [26], and history data could be easily asked from patients by predefined questions.

Three non-invasive features were highlighted by the analyses of this study; $SPO_2$, age, and the presence of cardiovascular disorders. Previous studies have shown that older age is positively associated with increased mortality in hospitalized COVID-19 patients [27]. Older age is associated with more infection susceptibility and an atypical response to viral pathogens due to reduced expression of type I interferon-beta [28]. Furthermore, age-related impairment of lymphocyte function along with an abnormal expression of type 2 cytokines leads to prolonged pro-inflammatory responses; this weakens the host response to viral replication causing poor clinical outcomes and higher mortality [29]. In contrast to typical types of pneumonia, the initial phases of COVID-19 have little apparent symptoms, such as dyspnea. The cause is the fact that there is still carbon dioxide exchange through alveoli at these stages. However, the oxygen exchange is disturbed due to the alveolar collapse. This type of hypoxia, called "silent hypoxia," leads to the progression of pneumonia in the absence of clinical symptoms [30]. It also causes a vicious cycle, where hypoxia promotes the activity of the local inflammatory system causing further damage and higher hypoxia [31]. Moreover, activation of the hypoxia-inducible factor-

$1\alpha$ could facilitate the ignition of cytokine storm in hypoxic patients through promoting proliferation in inflammatory cells and activation of pro-inflammatory cytokines [32]. therefore, $SPO_2$ could be a decisive factor to uncover the pneumonia progression and the severity state of patients. Pulse oximetry, via wearable devices or hospital equipment, could show decreased levels of $SPO_2$, in the blood; this is valuable for early detection of hypoxemia. Several studies have demonstrated that preexisting cardiovascular diseases, including acute coronary syndrome, arrhythmia, and heart failure, can worsen the outcome of COVID-19 patients. Furthermore, the SARS-CoV-2 virus exacerbates cardiac damages through direct interaction with ACE2 receptors [33].

In this study, LDH, PTT, and BUN had the highest mortality prediction weights among laboratory features. Elevated levels of LDH could reflect tissue injury caused by SARS-CoV-2 and concurrent lung fibrosis. Indeed, abnormal LDH is commonly seen in idiopathic lung fibrosis [34]. During the course of COVID-19, higher levels of LDH were observed during both alveolitis and fibrosis stages [35], highlighting it as a candidate for predicting the need for invasive ventilation [36] and mortality. In addition, a robust immune response to SARS-CoV-2 infection and subsequent cytokine storm could cause multi-organ damage, which leads to further elevation of LDH levels [37]. The inflammatory response promoted by severe SARS-CoV-2 infection could cause endothelial damage, distortion of the coagulation cascade function, and coagulopathy. Therefore, levels of PTT, a coagulation biomarker, during COVID-19 infection can be informative of coagulopathy progression and disease severity [38]. Several studies have found BUN as a predictor for adverse outcomes in COVID-19 patients [39]. A multicenter retrospective cohort conducted on 12,000 Chinese patients demonstrated that, even on the first day of admission, the BUN levels of patients who later expired were significantly higher than those who survived. Moreover, the dynamics of changes in BUN levels were quite different between expired patients and survivors within twenty-eight days of admission. Although the exact mechanism of the increase in BUN levels in COVID-19 patients is fully unraveled, several potential mechanisms have been proposed. The primary receptor of SARS-CoV2 is ACE2, which is highly expressed in epithelial cells in the kidney. Therefore, the SARS-CoV2 can activate the RAAS system through interaction with ACE-2 receptors. The consequent increase in the resorption of urea in renal tubules leads to a rise in BUN levels. Furthermore, another cause of the observed rise in BUN levels could be due to increased activity of inflammatory factors and cells, such as neutrophils, lymphocytes, and cytokines, which can systematically damage kidney tissue and alter renal function [40].

## Limitations

The results of this study should be interpreted in light of several limitations. This study was carried out within a retrospective framework. Consequently, supervision was not possible to increase the quality of data documentation when patients were admitted. Furthermore, the data gathering interval of this study encompassed the first pandemic wave, and medical records were documented in haste as high patient loads and limited medical staff forced the medical system to prioritize patient treatment. Therefore, many patients had incomplete medical profiles and were sieved before the data inspection phase. The aforementioned factors limited the sample size of the study. Researchers were not blind to outcomes. No external validation data was utilized due to limits imposed by the pandemic state of hospitals and preventive regimes. The Massih Daneshvari Hospital had more severe and expired patients since it was a primary care center for COVID-19. Thus, the severity and mortality rates of this study do not reflect the population rates of these variables. This could add confounding effects to the study. Finally, qualitative CRP, a feature reported by several studies to be associated with disease

severity, was removed from the analysis due to high missing values owing to limited laboratory resources and incomplete medical records caused by the pandemic. However, with the presence of other acute phase reactants and inflammatory markers, such as ESR, platelet number, and LDH, in this model, it is likely that a significant portion of variance explained by CRP was compensated by these features.

## Future works

To increase speed and convenience, imaging features were not utilized in this study. Future researches can compare the prediction power of imaging features with laboratory and non-invasive features. This study, conducted as a pilot study, was not externally validated. Future studies could include data from other hospitals for external validation. Furthermore, larger and more diverse study populations can be used for further evaluation of our results. Future projects can expand the practicality of our study by devising prognosis prediction software on various platforms. In this study, binary outcome (i.e., discharged and expired) was used as outcome. prospective projects can focus on other outcomes, such as whether a patient was intubated or admitted to ICU as outcomes. To devise specific prognostic models, Future studies can focus on individual groups of comorbidities (e.g., cardiovascular) and additional features to develop separate models. Continuous data input from various hospitals could be used to develop and incrementally train an online learning model to predict the prognosis of COVID-19 patients, giving increasingly precise and updated results to be used in clinical and non-clinical settings. Finally, the framework of this study could be tested in other acute respiratory infectious diseases to investigate the feasibility of mortality prediction via ML modeling of non-invasive features.

## Conclusion

Prediction of mortality prognosis during the COVID-19 pandemic is an important concept that can reduce disease mortality rates by giving us insights into where and with whom to intervene. The prognostic prediction capacity of laboratory biomarkers is distinct from those of clinical and demographic data. To investigate these differences in this study, predictor features obtained from patients' first day of admission were divided into invasive laboratory tests and non-invasive demographic and clinical data. Three prognostic machine learning models were developed using the aforementioned invasive and noninvasive biomarker groups; two using each of these groups and one using both. The models displayed optimal prediction performance, making them valuable assistive tools in different settings for clinical decision making and resource allocation. Furthermore, the implemented non-invasive model can be used for rapid triage of patients without the need for additional costs or waiting time of laboratory or imaging tests.

Analysis of invasive and non-invasive models across feature numbers revealed that the non-invasive model provided better performances in sparse dimensions with lower feature numbers. This points to the presence of a significant concentration of information in several non-invasive features, such as $SPO_2$ and age, and adds weight to the hypothesis that, given enough features, the information content of noninvasive factors could provide mortality predictions which are as good as high dimensional laboratory models. Temporal analysis across patient expiration intervals revealed that while invasive biomarkers are better predictors for near future deaths, they were outperformed by non-invasive biomarkers over larger expiration temporal intervals. This study aimed to explore the concept of combining machine learning techniques with non-invasive features towards the ultimate goal of development of rapid, automated triage tools; future studies should further explore this concept across different disease, feature, and population spectra.

## Methods

To increase the reporting quality and clarification, we aimed to follow the Transparent Reporting of a multivariable prediction model for individual prognosis or diagnosis (TRIPOD) reporting style to the extent that pandemic situational limitations permitted.

### Ethics

The protocol for this study was approved by the Ethics Committee of Shahid Beheshti University of Medical Sciences (SBMU). Written informed consent was obtained from all patients and, if applicable, their legal guardians regarding patient data utilization for medical research upon admission. This study was carried out in accordance with the Helsinki and SBMU guidelines and regulations. Patient data were anonymized before the data analysis phase.

### Study setting and population

This retrospective study was carried out using archived electronic medical records of COVID-19 patients who were admitted to the Masih Daneshvari Hospital in Tehran, Iran. As Iran's largest respiratory and pulmonary care center, the Masih Daneshvari Hospital was one of the first medical centers that admitted COVID-19 patients early during the pandemic.

The inclusion criterion for this study was hospital admission due to the initial diagnosis of COVID-19 infection by a physician according to the 5th Iranian COVID-19 guideline (Table 1). To enter this study, 628 patients who were admitted between February 20th, 2020, to May 4th, 2020, were randomly selected from the hospital's patient data pool. The review of medical records was initiated on June 5th. Medical records were reviewed separately by two physicians. Two pulmonologists adjudicated discordances. Afterwards, 12 patients who left the hospital against medical advice with consent, 27 patients with uncertain or rolled out COVID-19 diagnosis, 29 patients who were referred from other hospitals, 4 patients with age under 18, 43 patients with more than 20% missing data, 18 patients who received radically different treatment protocols (i.e., were enrolled in clinical trials), 1 patient who had a cardiac arrest shortly after arrival to the emergency ward, and 2 pregnant patients were excluded. Afterwards, 186 patients with an "expired" outcome (37.8%) and 306 patients with "discharged" outcome (62.2%) were included in the study (S1 Fig in S1 File).

### Definition of variables

Data from the first 24 hours of patients' admission was used in this study. The initial data, comprised of 57 features, was categorized into two groups of features; demographic and patient history features were labeled as non-invasive group, and laboratory results were labeled as invasive group. Demographic and history features were extracted from admission history, medical progress notes, and nursing notes. Laboratory features were extracted from the results of the first blood tests, which were ordered by physicians during the initial 24 hours of admission. Features with more than 10% missing values were entirely omitted. Imputation via the k-nearest neighbor (KNN) algorithm with k = 5 and uniform weights was used for features with less than 10% missing values; KNN algorithm imputes every missing value using the mean value from 'k' closest data points found in the training set.

The severity of a patient's condition was evaluated by physicians upon admission according to the criteria presented in Table 1. With the large magnitude of the outbreak and limited ICU beds, the severity criteria were tuned to the hospital's patient load and equipment to provide an efficient containment response. The blood pressure of patients was measured via electronic blood pressure patient monitoring devices by a physician or nurse. The oxygen saturation of

patients was measured in the room air (i.e., without oxygen support) using the hospital's pulse oximeters. The pulse rate of patients was measured using hospital pulse oximeters. The temperature of a patient was measured using digital forehead thermometers. Vital signs used in this study were those that were the first measurements of these features upon admission. The presence of a history of hypertension (HTN) was defined as the diagnosis of hypertension for the patient by a physician. The presence of a history of diabetes mellitus (DM) was defined as a diagnosis of DM type I or II for the patient by a physician. The presence of a history of cardiovascular diseases for the patient was defined as a history of Ischemic Heart Disease (IHD), Acute Coronary Syndrome (ACS), and Heart Failure (HF) that was diagnosed by a physician. Blood samples were obtained from venous blood and analyzed in the central medical laboratory of the Masih Daneshvari Hospital. S1 Table in S1 File contains a list of model input features along with their definitions.

To predict mortality outcomes, the framework of this study was expressed as a classification problem. Two outcome classes were defined; the discharged group (outcome = 1) consisted of COVID-19 patients that were discharged after the completion of their treatment and two consecutive negative PCR results. The expired group (outcome = 0) consisted of patients who died at any point during their treatment course.

## Statistical tests

The Kolmogorov–Smirnov test was used to examine distribution normality. Mean and standard deviation were used to describe normal continuous variables. The median and interquartile range were used to describe non-normal Continuous variables. Categorical data were expressed as frequency in percent. Mann Whitney U and Fisher's exact tests were used to test significance for numerical and categorical variables, respectively. Standardization (Z-score normalization) was used for feature scaling of non-categorical features. Initial statistical analysis was performed using SPSS 26.0 (IBM Corp. Released 2019. IBM SPSS Statistics for Windows, Version 26.0. Armonk, NY: IBM Corp) with P-values significance threshold of 0.05.

## Support Vector Machine analysis

Machine learning analyses were implemented using MATLAB version R2019b and the Statistics and Machine Learning Toolbox$^{TM}$ package [41].

Support Vector Machine (SVM) classifiers were developed based on demographic (2 feature), laboratory (26 feature), and clinical (9 feature) information (Fig 2A) (S1 Table in S1 File). Although a simple model, SVM frameworks display strong performance on small and medium-sized tabular datasets [42]. The SVM model, especially with a linear kernel, is self-explainable; we can inspect feature importance by looking at their weights [43]. This property facilitates the inspection and interpretation of feature contributions towards the outcome. The SVM binary classification algorithm searches for an optimal hyperplane that separates the data into two classes. For separable classes, the optimal hyperplane maximizes a margin (space that does not contain any observations) surrounding itself, which creates boundaries for the positive and negative classes. For inseparable classes, the objective is the same, but the algorithm imposes a penalty on the length of the margin for every observation that is on the wrong side of its class boundary.

For the implementation of the main SVM model analyses, the data were first divided into training and test sets with a holdout ratio of 0.2 using MATLAB's *cvpartition* function. Afterwards, model training and tuning were carried out using MATLAB's *fitcsvm* function and Sequential Minimal Optimization (SMO) was utilized as the solver. Kernel scale hyperparameter tuning was carried out using the *HyperparameterOptimization* input of the *fitcsvm*

command on the training data over 30 evaluation iterations via the function's default bayesian optimization. Kfold cross-validation with 10 folds was utilized for model training and hyper-parameter tuning via the *KFold* argument of the *fitcsvm* function. MATLAB command *kfold-Predict* was used to obtain labels and scores from the cross-validated model for performance assessment; for each fold, this command provides prediction labels and scores for in-fold instances using a model trained on out-of-fold instances and concatenates them. Parameters for the ROC curve and AUC measurement were obtained using the *perfcurve* MATLAB function and scores from the cross-validated model. The accuracy of the optimized cross-validated model was evaluated on the test set. The mean and standard deviation of performance metrics was calculated over 20 iterations of training and evaluation.

## Neighborhood Component Analysis

To investigate the principal features for mortality prediction, regularized Neighborhood Component Analysis (NCA) was utilized. This non-parametric analysis is an embedded method for selecting features with the goal of maximizing the prediction accuracy of regression and classification algorithms. The framework tries to learn feature weights for minimizing an objective function that measures the average leave-one-out classification or regression loss over the input data [44]. Function *fscnca* from MATLAB was utilized to perform NCA analysis with regularization. To obtain feature weights in Fig 2A, 200 iterations of NCA training were utilized. In each iteration, NCA training was implemented using all the data instances in 10 folds (*NumPartitions* input of the *fitcnca* function was set to 10). Limited memory Broyden-Fletcher-Goldfarb Shanno (LBFGS) algorithm was used as the solver. Regularization was implemented through the *Lambda* input of the *fscnca* function to decrease model variance and stabilize outputs. NCA weights were averaged over all the folds for each iteration then overall iterations to obtain final results (Fig 2A).

## Recursive feature elimination via SVM

Support Vector Machine-recursive feature elimination (SVM-RFE) is a feature evaluation method that utilizes the feature weights from the SVM algorithm to eliminate features with low importance in a recursive manner [45]. In this study, SVM-RFE was implemented using custom MATLAB code over 40 iterations. In each iteration, a linear support vector machine was trained without optimization overall data instances and 10 folds using *fitcsvm* function and its *KFold* input. Afterwards, accuracy was calculated using predictions that were obtained using the *kfoldPredict* method of the trained algorithm. This method returns concatenated labels of predictions over 10 folds; for each in-fold instance, the prediction is carried out using the algorithm trained on all the out-of-fold instances. In the next step, the feature with the lowest absolute weight was removed from the input dataset and the algorithm was retrained with new, reduced features and new partitioning. The aforementioned steps were continued until only one feature was left inside the input dataset. The mean and standard deviation of the accuracy was calculated over all iterations.

## Sparsity analysis

A combination of sparsity analysis with linear SVM (i.e., Sparse Linear SVM) is utilized for identifying the importance of feature subsets and evaluating their relevance in a computationally efficient framework [45]. In this study, we combined the least absolute shrinkage and selection operator (Lasso), a sparse regularization framework, with liner SVM to evaluate the predictive information content of invasive and non-invasive features with respect to the outcome. A custom code was written in MATLAB to implement 100 iterations of Sparse Linear

SVM analysis. In each iteration, 25 fixed, logarithmically space lambda values were utilized. For each lambda value, a linear SVM model without optimization was trained using *fitclinear* function utilizing all data instances as input and 10 folds (*KFold* input of the *fitclinear* was set to 10). The *Regularization* input was set to *lasso*. Classification loss was obtained using the *kfoldLoss* method of the trained algorithm which returns the loss averaged over all the folds. The Number of non-zero features for each lambda was calculated as the total number of features whose SVM weights were not reduced to zero. The mode of the number of non-zero features was calculated for each lambda over all iterations and first instances were selected (e.g., if the modes were [8 8 7 6 6 5] for 6 lambda columns, then the first 8 and 6 columns were selected as representatives of non-zero feature numbers ([8 7 6 5]) of the column). The corresponding mean and standard deviation of the loss of the selected non-zero feature numbers were calculated using instances whose number of non-zero features was equal to the calculated mode in the corresponding lambda column.

## Supporting information

**S1 File.**
(PDF)

## Acknowledgments

The authors would like to thank Mr. Akbari for his assistance with data gathering. We would also like to show our gratitude to all healthcare professionals worldwide who are at the front-line especially all physicians, nurses, and personnel of the Masih Daneshvari Hospital for their contribution in the fight against COVID-19.

## Author Contributions

**Conceptualization:** Mahdi Mahdavi, Hadi Choubdar, Erfan Zabeh, Ehsan Kamrani.

**Data curation:** Mahdi Mahdavi, Hadi Choubdar, Michael Rieder, Atefeh Abedini, Arda Kiani.

**Formal analysis:** Mahdi Mahdavi, Hadi Choubdar, Erfan Zabeh, Arda Kiani.

**Funding acquisition:** Michael Rieder, Safieddin Safavi-Naeini, Ehsan Kamrani.

**Investigation:** Mahdi Mahdavi, Hadi Choubdar, Erfan Zabeh, Michael Rieder, Safieddin Safavi-Naeini, Atefeh Abedini, Arda Kiani, Vida Khanlarzadeh, Reza Lashgari, Ehsan Kamrani.

**Methodology:** Mahdi Mahdavi, Hadi Choubdar, Erfan Zabeh, Reza Lashgari, Ehsan Kamrani.

**Project administration:** Ehsan Kamrani.

**Resources:** Michael Rieder, Safieddin Safavi-Naeini, Atefeh Abedini, Reza Lashgari, Ehsan Kamrani.

**Software:** Erfan Zabeh.

**Supervision:** Reza Lashgari, Ehsan Kamrani.

**Validation:** Mahdi Mahdavi, Hadi Choubdar, Erfan Zabeh, Michael Rieder, Safieddin Safavi-Naeini, Zsolt Jobbagy, Amirata Ghorbani, Atefeh Abedini, Arda Kiani, Vida Khanlarzadeh, Reza Lashgari, Ehsan Kamrani.

**Visualization:** Mahdi Mahdavi, Hadi Choubdar, Erfan Zabeh, Michael Rieder, Safieddin Safavi-Naeini, Zsolt Jobbagy, Amirata Ghorbani, Atefeh Abedini, Arda Kiani, Vida Khanlarzadeh, Reza Lashgari, Ehsan Kamrani.

**Writing – original draft:** Mahdi Mahdavi, Hadi Choubdar, Erfan Zabeh, Reza Lashgari, Ehsan Kamrani.

**Writing – review & editing:** Mahdi Mahdavi, Hadi Choubdar, Erfan Zabeh, Michael Rieder, Safieddin Safavi-Naeini, Zsolt Jobbagy, Amirata Ghorbani, Atefeh Abedini, Arda Kiani, Vida Khanlarzadeh, Reza Lashgari, Ehsan Kamrani.

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
