## [Decision Letter · Decision Letter 0]

22 Apr 2021

PONE-D-21-09671

A machine learning based exploration of COVID-19 mortality risk

PLOS ONE

Dear Dr. Kamrani,

Thank you for submitting your manuscript to PLOS ONE. After careful consideration, we feel that it has merit but does not fully meet PLOS ONE’s publication criteria as it currently stands. Therefore, we invite you to submit a revised version of the manuscript that addresses the points raised during the review process.

 Authors are to consider these models for developing countries? 

We look forward to receiving your revised manuscript.

Kind regards,

Usman Qamar

Academic Editor

PLOS ONE

Journal Requirements:

3. Please amend your Ethics Statement and Methods section to state the type of informed consent provided (e.g., written, verbal, etc).

5. Please include captions for your Supporting Information files at the end of your manuscript, and update any in-text citations to match accordingly. Please see our Supporting Information guidelines for more information: http://journals.plos.org/plosone/s/supporting-information

Reviewers' comments:

Reviewer's Responses to Questions

**Comments to the Author**

1. Is the manuscript technically sound, and do the data support the conclusions?

Reviewer #1: Yes

2. Has the statistical analysis been performed appropriately and rigorously? 

Reviewer #1: Yes

3. Have the authors made all data underlying the findings in their manuscript fully available?

Reviewer #1: Yes

4. Is the manuscript presented in an intelligible fashion and written in standard English?

Reviewer #1: Yes

5. Review Comments to the Author

Reviewer #1: I am profoundly interested in this research because really these models can create powerful frameworks for various medical assignments and patient triage. But do you considered these models for developing countries?

6. PLOS authors have the option to publish the peer review history of their article (what does this mean?). If published, this will include your full peer review and any attached files.

Reviewer #1: **Yes: **Thomas Ayalew Abebe

---

## [Author Response · Author response to Decision Letter 0]

12 May 2021

Dear Dr. Usman Qamar,

Academic Editor

PLOS ONE

Thank you for giving us the opportunity to resubmit our revised manuscript “A machine learning based exploration of COVID-19 mortality risk”, Ms. No: PONE-D-21-09671 to PLOS ONE. In this version of the manuscript, we have addressed all reviewer comments and made changes throughout the main manuscript text. We appreciate the time and effort that you and the reviewers have dedicated to providing your feedback on our manuscript. Below, we detail how we have responded to the reviewers’ comments. 

Sincerely,

Ehsan Kamrani, Ph.D

Reza Lashgari, Ph.D 

Reviewer Points:

- Authors are to consider these models for developing countries? 

Answer: Yes. The aim of this study is to pave the way for the implementation of computer-assisted triage in Iranian hospitals. As a developing country, Iran faced significant waves of patients that severely strained the healthcare system, causing a shortage of medical personnel and equipment. This study was conducted on Iranian patients in Masih Daneshvari Hospital, and the authors hope that by providing evidence for the performance of automated triage models in the Iranian population, healthcare systems of Iran and other developing countries be encouraged to support research, designation, and implementation of these models in medial environments to assist medical triage and decision making.

Editorial comments:

Answer: The manuscript was revised to meet the style requirements of the journal.

Answer: Reference list was reviewed to check for quality. Several references were updated to be more complete.

3. Please amend your Ethics Statement and Methods section to state the type of informed consent provided (e.g., written, verbal, etc).

Answer: The ethics statement was amended to address this issue.

4.We note that you have indicated that data from this study are available upon request. PLOS only allows data to be available upon request if there are legal or ethical restrictions on sharing data publicly.

Answer: After consulting with the hospital’s committee of medical research, permission was granted for sharing of patient anonymized data. The url for the dataset is included in the cover letter.

5. Please include captions for your Supporting Information files at the end of your manuscript, and update any in-text citations to match accordingly

Answer: Captions for supporting information files were added to the end of manuscript and text was updated to match citations.

---

## [Editor Report · Decision Letter 1]

17 May 2021

A machine learning based exploration of COVID-19 mortality risk

PONE-D-21-09671R1

Dear Dr. Kamrani,

We’re pleased to inform you that your manuscript has been judged scientifically suitable for publication and will be formally accepted for publication once it meets all outstanding technical requirements.

Kind regards,

Usman Qamar

Academic Editor

PLOS ONE
---

## [Editor Report · Acceptance letter]

8 Jun 2021

PONE-D-21-09671R1 

A Machine Learning Based Exploration of COVID-19 Mortality Risk 

Dear Dr. Kamrani:

I'm pleased to inform you that your manuscript has been deemed suitable for publication in PLOS ONE. Congratulations! Your manuscript is now with our production department. 

Kind regards, 

on behalf of

Dr. Usman Qamar 

Academic Editor

PLOS ONE